# Nutritional Support in Acute Liver Failure

**DOI:** 10.3390/diseases10040108

**Published:** 2022-11-18

**Authors:** Ludovico Abenavoli, Valentina Maurizi, Luigi Boccuto, Arianna Di Berardino, Nena Giostra, Pierangelo Santori, Maria Laura Scarcella, Anna Caterina Procopio, Carlo Rasetti, Emidio Scarpellini

**Affiliations:** 1Department of Health Sciences, University “Magna Græcia”, 88100 Catanzaro, Italy; 2Internal Medicine Residency Program, Università Politecnica delle Marche, 60121 Ancona, Italy; 3Healthcare Genetics and Genomics Doctoral Program, School of Nursing, College of Behavioral, Social and Health Sciences, Clemson University, Clemson, SC 29634, USA; 4Clinical Nutrition and Internal Medicine Unit, “Madonna del Soccorso” General Hospital, 63074 San Benedetto del Tronto, Italy; 5Hepatology and Internal Medicine Unit, “Madonna del Soccorso” General Hospital, 63074 San Benedetto del Tronto, Italy; 6Anesthesia, Intensive Care and Nutritional Science, Azienda Ospedaliera “Santa Maria”, Via Tristano di Joannuccio, 05100 Terni, Italy; 7T.A.R.G.I.D., Gasthuisberg University Hospital, KU Leuven, Herestraat 49, 3000 Leuven, Belgium

**Keywords:** liver failure, nutrition, branched-chain amino acids, gut microbiota, dysbiosis, probiotics

## Abstract

Acute liver failure (ALF) presents with an acute abnormality of liver blood tests in an individual without underlying chronic liver disease. The clinical course leads to the development of coagulopathy and hepatic encephalopathy. The role of nutrition in its prevention and treatment remains uncertain. We aimed to review literature data on the concept of ALF and the role of nutrition in its treatment and prevention, considering the impact of gut microbiota dysbiosis and eubiosis. We conducted a review of the literature on the main medical databases using the following keywords and acronyms and their associations: liver failure, nutrition, branched-chain amino acids, gut microbiota, dysbiosis, and probiotics. Upon their arrival at the emergency department, an early, accurate nutritional assessment is crucial for individuals with ALF. Branched-chain amino acids (BCAAs), stable euglycemia maintenance, and moderate caloric support are crucial for this subset of patients. An excessive protein load must be avoided because it worsens hepatic encephalopathy. Preclinical evidence supports future probiotics use for ALF treatment/prevention. Nutritional support and treatment for ALF are crucial steps against patient morbidity and mortality. BCAAs and euglycemia remain the mainstay of nutritional treatment of ALF. Gut dysbiosis re-modulation has an emerging and natural-history changing impact on ALF.

## 1. Introduction

Acute liver failure (ALF) is a rare and challenging condition involving very often the emergency department (ER) team and, therefore, either the intensive care unit (ICU) or the mild-intensity hepatology clinic [1]. Typically, patients present with unexplained symptoms/signs of hepatic encephalopathy and coagulopathy in the blood test analyses [2]. It is also worth mentioning that the patients can be adults or children [3]. Importantly, these patients presenting with desperate clinical conditions do not have a pre-existing liver disease and/or cirrhosis [4,5]. Anamnestic investigation can help physicians recognize the aetiologic agent of acute liver injury (ALI) [6]. For example, the use and—very frequently—abuse of medications such as the over-the-counter acetaminophen, whose cut-off dosage is narrow and variable among people, is one of the most common etiologies of ALI leading to ALF. The prognosis of ALF can be extremely poor and require liver transplantation [7,8]. Malnutrition is a common condition in ALF and is an independent prognostic factor for morbidity and mortality. The deficiency of vitamins, minerals, glucose, and amino acids must be assessed and eventually addressed very early in the frame of patient management [9]. However, liver inflammation and consequent destruction open up the tight junctions maintaining a “vigil” intestinal permeability and release damage-associated molecular patterns (DAMPs), such as Gram-negative lipopolysaccharide, which can pass through the portal vein and perpetuate liver damage until necrosis via oxidative stress. This is an example of a disrupted gut–liver axis and is the pathophysiological loop explaining the pathogenesis of several chronic liver diseases such as liver steatosis (NAFLD), non-alcoholic steatohepatitis (NASH), liver cirrhosis, and hepatocellular carcinoma [10]. Within this pathophysiological model, an emerging actor is gaining increasing attention among researchers: gut microbiota, whose qualitative and quantitative composition is crucial for host homeostasis [11]. Its dysbiosis is responsible for several pathologic conditions involving liver inflammation and fibrosis, including ALF. For these reasons, we aimed to review literature data regarding the definition of ALF and the role of nutrition in its treatment and prevention, with a special focus on gut microbiota dysbiosis and its pathophysiological impact and potential therapeutic application.

## 2. Materials and Methods

We conducted a PubMed and Medline search for original articles, reviews, meta-analyses, and case series using the following keywords, their acronyms, and associations: liver failure, nutrition, branched-chain amino acids, gut microbiota, dysbiosis, and probiotics. When appropriate, we also included preliminary evidence from abstracts belonging to main national and international gastroenterological meetings. The papers found from the above-mentioned sources were reviewed by two of the authors (ES and VM) according to PRISMA guidelines [12]. The last Medline search was conducted on 30 June 2022.

In detail, we found 262 manuscripts matching our search: 52 were clinical trials/original articles, and of these, 24 were randomized clinical trials (RCT)s, in addition to 107 reviews of the scientific literature, 20 of them systematic reviews and/or metanalyses. In detail, we included data from 74 papers: 20 original articles (including in vitro and animal studies); 12 RCTs; 11 systematic reviews of the scientific literature, excluding those not updated; and 9 metanalyses.

## 3. Results

### 3.1. Acute Liver Failure: Definition and Clinical “Scenarios”

ALF is a rare syndrome characterized by an acute abnormality of liver blood tests, due to a primary liver insult, in subjects without underlying chronic liver disease. ALF’s main features are coagulopathy and hepatic encephalopathy. Of importance, patients who develop coagulopathy without encephalopathy have an ALI [13]. From a wider point of view, ALF clinical presentation includes non-specific mild symptoms (e.g., fatigue, malaise, anorexia, nausea, and/or vomiting) and, rising in gravity, abdominal pain, jaundice and itch, ascites, mental confusion, agitation, and coma because of cerebral edema. The development of multiorgan failure (MOF) is not uncommon. In detail, hepatic encephalopathy manifestations range from changes in behavior to coma. They have five grades according to the West Haven classification [14]. At grade 0, patients have no changes in personality/behavior; there is just mild loss of memory, concentration, and coordination. Asterixis is the hallmark of grade I, together with deranged orientation and sleep disturbances (hypersomnia or insomnia); patients start to fail neuropsychological tests also because of decreased attention. These patients can also show mood alterations (both depression and euphoria). In grade II, lethargy and/or apathy are characteristic symptoms, although personality changes toward irritability can also be present. In grade III, severe confusion until reversible sleeping and gross disorientation are present. In the fourth grade, there is a coma [15,16]. Intriguingly, “overt” encephalopathy is another syndromic association, difficult to diagnose, because it is characterized by marked alteration of mental status, asterixis, hyperreflexia, and hypertonicity that can easily evolve to coma and exitus [3,4,5,17]. Importantly, since initial mental impairment may be subtle, the identification of hepatic encephalopathy in a patient without subsiding cirrhosis and/or preexisting liver disease requires an accurate differential diagnosis. Coagulopathy, a criterion required for diagnosis, is defined by a prolongation of prothrombin time or prolongation of the international normalized ratio (INR) >1,5 [18]. The incidence of ALF ranges from one case per million people per year to about 2000–3000 cases per year in the USA [19]. However, in developing countries, the incidence is significantly higher [20]. Acute liver failure can result from a wide variety of causes (Table 1).

Acute viral hepatic damage remains the most common cause of ALF worldwide. However, it has sensibly declined in the European Union because of direct antiviral drugs used for HCV-related infections and HBV large-scale vaccination. Thus, drug-induced liver injury (DILI) is the main cause of ALF (e.g., from acetaminophen intoxication) in European countries. Moreover, autoimmune hepatitis, Budd–Chiari syndrome, and/or Wilson disease may have an acute onset with coagulopathy and encephalopathy and should be considered as causes of ALF in the absence of overt fibrosis and/or cirrhosis [21]. From a clinical point of view, O’Grady classified ALF as hyperacute, acute, and subacute according to encephalopathy onset when it develops within 7 days, 8–28 days, or 5–26 weeks, respectively, upon jaundice appearance [22]. Intriguingly, this classification was built up to be associated with the prognostic state. However, the associations reflected mainly the underlying causes as determinants of patient outcomes. For example, hyperacute presentation is more often related to acetaminophen or ischemic liver damage. Both of these causes can result in cerebral edema and/or extrahepatic MOF. On the other hand, they have a good outcome after medical treatment vs. acute or subacute ALF. The latter have a poorer outcome without a liver transplantation attempt. Indeed, the survival rate of patients with ALF has dramatically improved in the last few decades: it has gone from 20% to more than 60%. This radical change can be explained by the combination of the emerging practice of multidisciplinary medical teams and by the growing use of emergency liver transplantation (ELT) when needed [23]. As the medical management of patients with ALF in the ER, ICU, and/or mild intensity unit also requires the early treatment of the underlying causes of this condition and general supportive measures, appropriate nutritional support is crucial for patients’ morbidity and mortality.

### 3.2. Nutritional Support in Acute Liver Failure

Recent ESPEN guidelines outlined a strong relationship between liver diseases and nutritional status [24]. In detail, in ALF patients there is a dramatic loss of hepatocellular function and subsequent MOF. From a metabolic point of view, there is a severe impairment of carbohydrate, protein, and lipid metabolism. In detail, these patients present with reduced glucose synthesis and lactate elimination by the liver, protein catabolism associated with hyperaminoacidemia, and hyperammonemia [9]. Malnutrition has a variable prevalence in ALF patients, also according to its hyperacute, acute, or subacute course. Indeed, it is an independent risk factor for morbidity and mortality, such as in chronic liver diseases. More in detail, typical factors of ALF malnutrition depend on fat malabsorption and impaired gastric emptying [25]. Nutritional support in liver failure should optimize caloric, macronutrient, and micronutrient supplementation. In addition, nutritional support should save the remaining liver function with improved metabolic reserve. Altogether, these can reduce the risk of infection and improve the overall performance of patients in general and of those candidates for liver transplantation in particular [10,26]. Sarcopenia can be prevented just after an accurate assessment of nutritional status (e.g., bioimpedance, indirect calorimetry): intake of calories, protein, and essential amino acids has to be assessed and has to be maintained to stabilize muscle mass. Thus, although branched-chain amino acid (BCAA) supplementation is crucial for the management of hepatic encephalopathy and sarcopenia, L-leucine has been described as the main amino acid involved in protein turnover and is able to reverse the proteolytic process towards protein synthesis [27]. L-leucine belongs to the BCAA group (valine, leucine, and isoleucine). Almost 40% of the amino acids needed by mammals are BCAAs. Moreover, these BCAAs altogether or leucine alone can sustain protein synthesis and can inhibit protein breakdown [28].

From a wider hepatologic perspective, in healthy people in general and in patients with liver cirrhosis specifically, BCAA supplementation promotes anabolic pathways and reduces cachexia, prevents and/or treats hepatic encephalopathy, alleviates fatigue during exercise, promotes wound repair, and stimulates insulin production [29,30]. Going in more detail, hydroxy-beta-methyl butyrate (HMB) is an active metabolite of leucine, with anti-catabolic properties, used in several randomized controlled trials (RCTs) in patients with liver cirrhosis against sarcopenia and fat accumulation in skeletal muscles [31,32,33]. In addition, a systematic review and consequent meta-analysis of RCTs showed HMB and/or nutritional supplements containing HMB to increase skeletal muscle mass and promote its strength and hypertrophy/hyperplasia [34]. From a molecular point of view, the interaction of leucine and HMB activates the mTOR metabolic pathway, increasing protein translation through the phosphorylation of eukaryotic initiation factor 4E binding protein 1 (4E-BP1) and ribosomal protein S6 kinase (S6K), reaching muscle hyperplasia [35,36,37].

Within a detailed ALF focus, according to the European Association for the Study of the Liver, recommended nutritional support in liver failure, regarding BCAAs, the suggested doses are L-leucine 2500 mg, L-isoleucine 1250 mg, L-valine 1250 mg, HMB 1500 mg (namely, the therapeutic dosage), vitamin D 40 mcg (therapeutic dosage) or 1600 I.U., and vitamin K 60 and 400 micrograms. For prebiotics and B-vitamin complex, the recommended doses are about 25 milligrams of vitamin B1, B2, B6, and niacin; 400 to 800 micrograms of folic acid; and 25 micrograms of vitamin B12.

Enteral feeding is the preferred choice vs. parenteral nutrition in these patients. Oral nutrition should be encouraged but low-consciousness status due to rapid progressive hepatic encephalopathy or anorexia may impair this method of nutrient administration. Thus, nasogastric tube insertion is recommended, considering the risk of aspiration. For example, if delayed gastric emptying is present, feeding tubes can be placed post-pylorically. In addition, if a small-bowel failure occurs (e.g., due to ileus mechanical obstruction or ischemia), there is a higher rate of gut bacterial translocation and sepsis. Both events can be reduced by enteral feeding. Similarly to other ICU patients, total parenteral nutrition initiation is not recommended 5–7 days after ICU admission [38,39]. In the rare cases in which the parenteral route of administration has to be used, lipid emulsions are safe. Interestingly, SMOF lipids (SMOF) are newer preparations containing ω-3 and medium-chain triglycerides with higher patient performance vs. soy-based lipids [40]. Indeed, fat metabolism is regulated by active mitochondrial function, avoiding their potential accumulation and further liver damage [41]. Unfortunately, solid evidence in patients with ALF is lacking. Lipidic metabolism (triglycerides target >3 mmol/L or 265 mg/dL) has to be controlled along with creatinine kinase [40,41]. Caloric goals for enteral feeding in patients with ALF are driven by increased resting energy expenditure (between 18% and 30%) [42]. Hypoglycemia is a recurrent condition in subjects with ALF, who may develop hepatic glycolysis and impaired gluconeogenesis. These processes are concomitant with a hyperinsulinemic state and rapidly consumed glycogen reserves [43]. Thus, accurate glycemia monitoring should be performed and the administration of glucose at 1.5–2 g/kg/day is recommended in the ICU. Furthermore, both enteral and parenteral nutrition prevent hypoglycemia [44]. In detail, enteral nutrition should be started as soon as possible in both pediatric and adult patients with ALF. However, severe shock and/or gut dysfunction suggest going for parenteral nutrition. On the other hand, hyperglycemia must be avoided, as it can favor intracranial hypertension (ICH). Optimal blood-glucose levels are between 150 and 180 mg/dL [45]. Interestingly, hypertonic sodium chloride can be used to reach hypernatremia (e.g., values between 145 and 155 mmEq/L) to further reduce the incidence of ICH and consequent cerebral edema [40]. It is worth considering potential thiamin and vitamin B12 deficiencies check in patients with ALF suspected of alcohol/drug abuse. Patients may have also fat-soluble vitamins (e.g., A, D, E, and K) deficiency. In particular, the prolongation of clotting tests (namely, prolonged INR, partial thromboplastin time, and abnormal factor V) can be a feature of vitamin K deficiency. However, we must note that patients with ALF have a more frequent procoagulant tendency, which results in a hypercoagulable state [46,47]. Thus, there is no recommendation for vitamin K level dosing in these patients. Vitamin D is not only crucial for bone integrity and the prevention of osteoporosis but also modulates intestinal permeability and inflammatory response to pathogens [48]. Therefore, it can be supplemented for patients with ALF with low levels of vitamin D [49]. In addition, zinc—a microelement crucial for our immune system homeostasis—is also a cofactor for ammonia conversion to urea. Thus, zinc deficiency is strictly linked to hepatic encephalopathy [50]. Conversely, patients with ALF derived from undiagnosed Wilson disease can present with copper-driven toxicity [51]. The hydration of individuals with ALF is important in terms of hemodynamic state stability. Indeed, the hemodynamic state of patients with ALF is very similar to that of cirrhotic and/or septic patients: there is low systemic vascular resistance. However, a subject with ALF has less pronounced portal hypertension compared to cirrhotic patients. Thus, there is an urgent need to maintain the adequate perfusion of the brain and kidney to avoid cerebral edema and renal failure [52]. In particular, more patients with ALF need to maintain a cerebral perfusion pressure of 60–80 mmHg through intravenous fluid and vasopressor administration. The final goal is to reach a mean arterial pressure of at least 75 mm Hg [45,53]. Renal replacement therapy is indicated to reverse intravascular volume overload, acidosis, and electrolyte abnormalities with a significant reduction in mortality [45,54].

### 3.3. Gut Microbiota and Acute Liver Failure: A Potential Biologic Weapon Reversing the Syndrome

The human intestine hosts over 100 trillion microbes, prevalently bacteria. There are also viruses, fungi, archaea, and protozoa [55]. Gut microbiota participates in nutrient absorption and fermentation, modulate intestinal permeability (IP), and are implicated in host metabolism (e.g., carbohydrates absorption and processing, proteins putrefaction, bile acids formation, insulin sensitivity) and the modulation of mucosal and systemic immunity [56].

In general, some evidence from the literature suggests the degree of acute liver damage is inversely correlated with gut microbiota dysbiosis reduction. The latter is directly correlated with liver inflammation via the development of a gut–liver axis through healthy and disease-affected life [57,58]. Furthermore, the gut microbiome contains genes encoding anti-inflammatory upstream signals typical of ALF. This results in the regulation of the gene expression of MYC activation during ALF and counteracts liver injury [59].

Data on gut dysbiosis related to ALF development can be extracted from several reports on ALI. In detail, changes in the intestinal microbial composition circadian rhythm can affect acetaminophen-induced ALI and related ALF [57]. Indeed, 16S rRNA gene sequencing showed rhythm daily changes to be associated with changes in gut microbiota abundance: the ratio of Firmicutes/Bacteroides was significantly reduced at night. Accordingly, acetaminophen hepatotoxicity was more severe at night vs. in the morning. This finding can be explained by the increased abundance of the 1-phenyl-1,2-propanedione (PPD) consuming glutathione (GSH) in hepatocytes. Further, the oral gavage of acetaminophen weakens the intestinal mucosal barrier function (increasing “intestinal permeability”), allowing abundant pathogenic antigens to enter the liver, with worsening liver inflammation and damage [60]. Thus, acute liver damage can lead to gut dysbiosis, and this, in turn, can start and endure the “gut–liver axis” through intestinal tight-junction dysregulation, impaired intestinal permeability, and the passage of PAMPs and DAMPs that hit hepatocytes. This vicious circle is virtually infinite [10,11]. On the other hand, we cannot exclude the possibility that gut dysbiosis can cause a predisposition to acute liver damage and subsequent ALF.

Indeed, a crucial recent work by Schneider and co-workers shed a clear light on the role of gut microbiota in chemical-induced ALI leading to ALF. Subjects with intestinal dysbiosis are more sensitive to acetaminophen ALI. Moreover, within a cohort of 500,000 participants in the British Biobank, proton pump inhibitors (PPI) or long-term antibiotic use were associated with gut dysbiosis whose risk of ALF was significantly higher than those with gut microbiota eubiosis. In parallel, *Nlrp6^−/−^* mice (an intestinal gut dysbiotic model) also showed that gut dysbiosis is associated with the same ALI vs. wild-type mice. Even more, interestingly, this hepatic damage was reproduced after fecal bacteria transplantation [61]. Several animal studies have shown that probiotic administration can improve chemical-induced liver injury. For example, *Lactobacillus salivarius LI01* and *Pediococcus pentosaceus LI05* significantly showed lower levels of transaminases in ALF rats. In addition, they can prevent jaundice occurrence and, interestingly, improve the histological abnormalities typical of ALF within the liver [62]. More, interestingly this probiotic mixture can finely change the composition of the cecal microbiome [63,64]. On the other hand, from a metabolic point of view, gut microbiota metabolites (e.g., 1-phenyl-1,2-propanedione) can also aggravate liver injury through the consumption of hepatic glutathione and ultimately facilitate hepatic oxidative damage [22]. More in detail, bile acids metabolically produced by gut microbiota (namely, hexanoic acid, trigeminal, 1-hexadecanol, campesterol, d-lactose, and lithocholic acid) show a significant alteration in ALF rats. Thus, they can pathologically affect the rate of liver damage (Table 2) [65].

## 4. Conclusions

Nutritional support is crucial in the prevention and management of ALF patients. Specifically, amino acid, mineral, vitamin, and glucose supplementation is necessary to reduce mortality and the grade of encephalopathy. Therefore, tailored enteral nutrition is advised for the period of hospitalization. Gut microbiota modulation is an appealing preventive and therapeutic option for ALF. This suggestion is based on the evidence of gut dysbiosis involvement in liver damage initiation and perpetuation via the altered gut–liver axis. Future RCTs are needed to confirm these in vitro and animal evidence.

## Figures and Tables

**Table 1 diseases-10-00108-t001:** Main causes of ALF.

Cause of ALF
Acetaminophen
Other drugs
Acute viral hepatitis
Acute-onset autoimmune hepatitis
Wilson disease
Acute ischemia
Budd–Chiari syndrome
Veno-occlusive disease
Acute fatty liver of pregnancy
Cancer liver infiltration
Hepatectomy
Toxins
Sepsis
Heat stroke
Hemophagocytic lymphohistiocytosis

**Table 2 diseases-10-00108-t002:** Preclinical use of probiotics in ALF models.

Probiotic Used and Dose	Animal Model	Liver Damage	Effects	Ref.
*Lactobacillus reuteri* DSM 17,938 (oral gavage with 3 × 10^9^ CFU daily for one week)	Sprague-Dawley rats	D-GaIN (1.1 g/kg of body weight) intraperitoneally injected	↓ dysbiosis; ↓ transcription of inflammatory factors in hepatocytes	[65]
*Lactobacillus rhamnosus* GG (oral gavage with 2 × 10^8^ CFU/100 μL HBSS daily for 2 weeks)	Germ-free C57BL/6 mice	Acetaminophen (300 mg/kg of body weight) oral gavage	Production of 5-MIAA with the activation of Nrf2 within the liver to protect against drug-induced oxidative stress	[66]
MegaSporeBioticTM (MSB) (orally 1 × 10^9^ CFU/rat daily for 12 days)	Charles River Wistar white male rats	Acetaminophen (2 g/kg of body weight) oral gavage	↓ pro-inflammatory cytokines production, ↓ hepatocyte necrosis	[67]
*Bifidobacterium adolescents* CGMCC15058 (gavage 3 × 10^9^ CFU/mL PBS daily for 2 weeks)	Germ-free Sprague–Dawley (SD) rats	D-GaIN (1.1 g/kg of body weight) intraperitoneal injection	↓ levels of mTOR and inflammatory cytokines; ↑ levels of anti-inflammatory cytokines (e.g., interleukin-10)	[68]
*Bacillus cereus* (gavage at 3 × 10^9^ CFU/mL PBS daily for 2 weeks)	Sprague–Dawley rats	D-GaIN (1.1 g/kg of body weight) intraperitoneal injection	↓ inflammatory response; improved intestinal permeability; re-establishing of eubiota	[69]
*Akkermansia muciniphila* (oral gavage 1.5 × 10^9^ CFU/200 μL PBS daily for 2 days)	C57BL/6 mice	Alcohol (6 g/kg of body weight) oral gavage	↓ hepatic injury, steatosis, and infiltration of MPO+ neutrophils	[70]
*Akkermansia muciniphila* (oral gavage 3 × 10^9^ CFU/200 μL PBS daily for 2 weeks)	C57BL/6 mice	Con A (15 mg/kg of body weight) injection through the tail vein	↓ inflammatory cytokines production, cytotoxic factors and hepatocellular necrosis; ↑ gut microbiota diversity	[71]
*Saccharomyces boulardii* (gavaged with 1 × 10^9^ CFU/mL for 4 weeks)	BALB/c mice	D-GaIN (200 mg/kg of body weight) intraperitoneally injected	↑ *Bacteroidetes* abundance; ↓ the abundance of *Firmicutes* and *Proteobacteria*	[72]
*Lactobacillus casei Zhang* (gavaged with 10^9^ CFU/daily for one month)	Wistar rats	LPS / D-GalN (50 μg/kg and 300 mg/kg of body weight, LPS and D-GalN, respectively) intraperitoneal injection	Modulation of the TLR-MAPK-PPAR-g pathways leading to ↓↓ pro-inflammatory cytokine production and hepatic inflammation	[73]
*Lactobacillus rhamnosus* GG (mixed with drinking water approximately 10^9^ CFU/daily for 5 days)	C57BL/6N mice	Alcohol (6 g/kg of body weight) oral gavage	Activating HIF signaling to: ↓ hepatic damage, impaired intestinal permeability, and levels of endotoxemia	[74]

D-GaIN, D-galactosamine; 5-MIAA, 5-methoxyindoleacetic acid; Con A, concanavalin A.

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
