# Peer review of "Nutritional Support in Acute Liver Failure"

_diseases, 2022, doi:10.3390/diseases10040108_

Round 1
Reviewer 1 Report
This is a meaningful article which reviewed the definition of ALF, and the role of nutrition in its treatment and prevention, with a special focus on gut microbiota dysbiosis.
However, it is more of a review than a systematic review. Systematic review and literature search strategy were mentioned many times in this paper, but these are not standardized writing methods of systematic review. For example, the author did not check all the articles and give the inclusion and exclusion criteria. In the analysis of the results, nutritional support for hepatitis, liver cirrhosis, and chronic liver failure were also confused with acute liver failure. In addition, the article lacked a more systematic and rich summary and analysis of the related research on the influence of gut microbiota dysbiosis and its pathophysiological impact and potential therapeutic application on acute liver failure.
Author Response
Reply: we thank the Reviewer for these comments.
We agree that this is not a systematic review of the literature. Thus, we have edited the text accordingly. We have also added inclusion and exclusion criteria for article selection.
We made the text more consistent with the distinction of nutritional support for ALF.
Finally, we have specified within the text the sections on the pathophysiological and therapeutic role of gut microbiota in ALF pathophysiology and treatment. However, there is no direct evidence describing both gut dysbiosis composition and its impact on the pathophysiology of ALF. We have specified this within the text.
Regards
Reviewer 2 Report
There's a very interesting review of the importance of nutrition in the evolution and outcome of acute liver failure (ALF). The work explores the interaction between the causes and consequences of ALF and how they can be avoided / improved or aggravated by nutrition. The authors present an exhaustive review of the pathophysiology of ALF and the mechanisms leading to known complications of this disease, i.e. hipoglicemia, cerebral edema, sarcopenia, etc.
Interestingly, the current knowledge on the influence of nutrition in gut microbiota and, moreover, how this microbiota can modulate the clinical manifestations of ALF is extensively discussed.
I think this is an important review that can be of interest for physicians dealing with acute liver diseases.
Author Response
Referee#2
There's a very interesting review of the importance of nutrition in the evolution and outcome of acute liver failure (ALF). The work explores the interaction between the causes and consequences of ALF and how they can be avoided / improved or aggravated by nutrition. The authors present an exhaustive review of the pathophysiology of ALF and the mechanisms leading to known complications of this disease, i.e., hypoglycemia, cerebral edema, sarcopenia, etc.
Interestingly, the current knowledge on the influence of nutrition in gut microbiota and, moreover, how this microbiota can modulate the clinical manifestations of ALF is extensively discussed.
I think this is an important review that can be of interest for physicians dealing with acute liver diseases.
Reply: we thank the Reviewer for all the positive comments. We would like to respectfully suggest that perhaps ratings marked in the evaluation form could have been wrongly ticked up by the reviewer. Indeed, they are not aligned with all the positive comments made in this exhaustive manuscript evaluation. Thus, we thank once again the reviewer for his full appreciation of our work.
Regards
Reviewer 3 Report
This is review paper. The authors collected suitable paper to discuss. Then it is now text book.
Author Response
Referee#3
This is a review paper. The authors collected suitable paper to discuss. Then it is now text book.
Reply: we thank the Reviewer for these comments.
Regards
Reviewer 4 Report
It is a correct revision about the parenteral revision of nutritional support in ALF. We have not complete evidence of the importance of it in prevention and treatment due the absence of sufficient clinical trials/studies.
Author Response
Reply: we thank the Reviewer for these comments. We agree that we have not yet complete evidence of the importance of parenteral nutrition in ALF treatment.
Regards